# LBMPL Vaccine Therapy Induces Progressive Organization of the Spleen Microarchitecture, Improved Th1 Adaptative Immune Response and Control of Parasitism in *Leishmania infantum* Naturally Infected Dogs

**DOI:** 10.3390/pathogens11090974

**Published:** 2022-08-26

**Authors:** Bruno Mendes Roatt, Jamille Mirelle de Oliveira Cardoso, Levi Eduardo Soares Reis, Gabriel José Lucas Moreira, Letícia Captein Gonçalves, Flávia de Souza Marques, Nádia das Dores Moreira, Paula Melo de Abreu Vieira, Rodrigo Dian de Oliveira Aguiar-Soares, Rodolfo Cordeiro Giunchetti, Alexandre Barbosa Reis

**Affiliations:** 1Laboratório de Imunopatologia, Núcleo de Pesquisas em Ciências Biológicas, Universidade Federal de Ouro Preto, Ouro Preto 35400-000, Minas Gerais, Brazil; 2Departamento de Ciências Biológicas, Instituto de Ciências Exatas e Biológicas, Universidade Federal de Ouro Preto, Ouro Preto 35400-000, Minas Gerais, Brazil; 3Grupo de Imunologia para de Medicina, Nutrição, Centro Universitário Presidente Tancredo de Almeida Neves—UNIPTAN, São João del Rei 36301-182, Minas Gerais, Brazil; 4Laboratório de Morfopatologia, Núcleo de Pesquisas em Ciências Biológicas, Universidade Federal de Ouro Preto, Ouro Preto 35400-000, Minas Gerais, Brazil; 5Faculdade de Ciências Médicas, Universidade do Estado do Rio de Janeiro, Rio de Janeiro 21941-902, Rio de Janeiro, Brazil; 6Laboratório de Biologia das Interações Celulares, Departamento de Morfologia, Universidade Federal de Minas Gerais, Belo Horizonte 31270-013, Minas Gerais, Brazil; 7Instituto Nacional de Ciência e Tecnologia em Doenças Tropicais, INCT-DT, Salvador 40015-970, Bahia, Brazil

**Keywords:** canine visceral leishmaniasis, *Leishmania infantum*, immunotherapy, LBMPL vaccine, spleen

## Abstract

The spleen plays a central role in human and canine visceral leishmaniasis, where the activation of the immune response occurs in one of the tissues where *Leishmania infantum* reproduces. Therefore, this organ is both a target to understand the mechanisms involved in the parasite control and a parameter for assessing the therapeutic response. In this sense, this study aimed to evaluate the main histological, immunological and parasitological aspects in the spleen of symptomatic dogs naturally infected by *L. infantum* treated with the therapeutic vaccine LBMPL. For this, dogs were divided into four groups: dogs uninfected and untreated (NI group); *L. infantum*-infected dogs that were not treated (INT group); *L. infantum*-infected dogs that received treatment only with monophosphoryl lipid A adjuvant (MPL group); and *L. infantum*-infected dogs that received treatment with the vaccine composed by *L. braziliensis* promastigote proteins associated with MPL adjuvant (LBMPL group). Ninety days after the therapeutics protocol, the dogs were euthanized and the spleen was collected for the proposed evaluations. Our results demonstrated a reduction of hyperplasia of red pulp and follicular area of white pulp, increased mRNA expression of IFN-γ, TNF-α, IL-12 and iNOS, and decreased IL-10 and TGF-β1, and intense reduction of splenic parasitism in dogs treated with the LBMPL vaccine. These results possibly suggest that the pro-inflammatory environment promoted the progressive organization of the splenic architecture favoring the cellular activation, with consequent parasite control. Along with previously obtained data, our results propose the LBMPL vaccine as a possible treatment strategy for canine visceral leishmaniasis (CVL).

## 1. Introduction

Zoonotic visceral leishmaniasis (VL) caused by *Leishmania infantum* is a vector-borne infectious disease in the Americas and Mediterranean basin [1]. Domestic dogs are considered the main urban and peri-urban reservoirs of VL, being important in the epidemiology of the disease [2,3]. 

The study of canine visceral leishmaniasis (CVL) can contribute to the understanding of human disease, since some clinical features, histopathological alterations and disease evolution bear similarities with the human form [4]. The main clinical signs of canine and human VL are lymphadenopathy, hepatomegaly, splenomegaly, anemia, low platelet and leukocytes counts, cutaneous pigmentation and weight loss [5,6]. Still, the clinical presentation depends on the immunocompetency of the host and varies from asymptomatic to severe forms [7]. In addition, a complex signaling network produced by macrophages and T and B cells is responsible for an inflammatory status that cannot achieve the control of parasite growth in the severe forms of the disease [8,9]. 

The bone marrow, liver and spleen are the most affected internal organs [10]. The spleen being responsible for immune surveillance against blood circulating pathogens, its role is crucial in the *Leishmania* infection progression [11]. The spleen can rapidly produce nonspecific polyreactive antibodies, efficiently containing infectious agents until the antigen-specific immune reaction occurs in the germinal center [10,11]. However, in severe cases of VL, a splenic architecture disorganization is observed with hyperplasia of red pulp and atrophy, and disruption of the white pulp [6,10,12]. These biological events are indicators of infection progress in dogs and humans [13,14]. Thus, the spleen is a target to understand the mechanisms involved in the inability to control parasite burden and response to the treatment in severe cases [15]. In this sense, immunotherapeutic strategies have been investigated against the CVL [16]. Roatt et al. [17] evaluated the therapeutic response of the LBMPL vaccine composed of *L. braziliensis* antigens associated with the adjuvant Monophosphoryl lipid A (MPL) in the treatment of CVL in symptomatic dogs naturally infected with *L. infantum.* The dogs showed control of bone marrow and skin parasitism and reduction in the intensity of clinical manifestations. In addition to that, the treated animals with the immunotherapeutic protocol demonstrated restoration and normalization of biochemical and hematological parameters. 

The immunotherapeutic approach employed by Roatt et al. [17] could boost the splenic immune response and promote a clinical improvement and efficient control of parasite burden in the tissue compartment. To confirm this hypothesis, the present study evaluated the histological, immunological and parasitological alterations in the spleen of symptomatic dogs naturally infected by *Leishmania infantum* treated with the therapeutic vaccine LBMPL.

## 2. Results

### 2.1. Histopathological Alterations Were Less Pronounced in the Splenic White and Red Pulp of Animals Treated with the LBMPL Vaccine

The analysis of hyperplasia of red pulp demonstrated a higher intensity (*p* < 0.05) in INT and MPL groups when compared to the NI group (Figure 1A). Moreover, we observed a lower intensity (*p* < 0.05) of red pulp hyperplasia in the LBMPL group compared to the INT and MPL groups (Figure 1A). In the representative photomicrographs, it is possible to observe the absence of red pulp hyperplasia in NI and LBMPL groups (Figure 1A). On the other hand, we observed intense hyperplasia of red pulp macrophages with loss of histological organization in INT and MPL groups, demonstrating little definition of the boundaries between the white and red pulp in all splenic tissue.

Based on white pulp analysis, a reduction (*p* < 0.05) in the follicular area was observed in the LBMPL group compared to the INT and MPL groups (Figure 1B). The histopathological findings demonstrated large lymphoid follicles and disorganized splenic white pulp in the INT and MPL groups (arrows). The white pulp was evident, but its regions were poorly defined or indistinct (Figure 1B). By contrast, smaller, and more organized follicles with a discernible germinal center, mantle zone and marginal zone were observed in NI and LBMPL groups (Figure 1B).

### 2.2. Subtype Th1 Cytokine Expression Is Higher in the Spleen of Dogs Treated with LBMPL Vaccine

IFN-γ and iNOS gene expression were higher (*p* < 0.05) in LBMPL groups than in INT and MPL groups (Figure 2). Higher gene expression of TNF-a (*p* < 0.05) was found in the splenic tissue of MPL and LBMPL groups than in the INT group (Figure 2). Furthermore, the mRNA expression of IL-12 was increased (*p* < 0.05) in the LBMPL group when compared to the INT group (Figure 2). Based on Th2 cytokines, a lower expression (*p* < 0.05) of IL-10 and TGF-β1 was observed in the LBMPL group compared to the INT group (Figure 2). 

### 2.3. LBMPL Vaccine Promoted a Strong Reduction in Splenic Parasite Burden in Treated Dogs

Our results demonstrated that the LBMPL vaccine promoted a significant reduction (*p* < 0.05) in splenic parasite burden when compared to the INT and MPL groups (Figure 3A). This reduction was approximately 96% when compared to the INT group. 

Correlation analyses were performed by Spearman’s r test between the spleen parasite load and cytokines/iNOS expression in this organ. The main results are shown in Figure 3B, where a negative linear (*r* = −0.859, *p* = 0.028) correlation can be seen between IFN-γ expression and parasite load, and a positive correlation (*r* = 0.601, *p* = 0.043) between IL-10 expression and parasite load in the INT group (Figure 3B). Moreover, we observed a negative correlation between IFN-γ (*r* = −0.686, *p* = 0.050), TNF-α (*r* = −0.743, *p* = 0.034) and iNOS (*r* = −0.770, *p* = 0.042) expression and parasite burden, and a positive correlation (*r* = 0.735, *p* = 0.025) between IL-10 expression and parasite load in the LBMPL group (Figure 3B). 

## 3. Discussion

The impact of therapeutic protocols should positively affect central organs such as the spleen, which controls the immune surveillance against pathogens. In this context, we performed a detailed study of the histopathological, immunological and parasitological aspects of the spleen of dogs naturally infected with *L. infantum* and submitted to immunotherapy with the LBMPL vaccine. Our main findings demonstrated a reduction in splenic red pulp hyperplasia and white pulp follicular area, increased and decreased RNA expression of Th1 subtype cytokines and Th2 subtype cytokines, respectively, and intense reduction in splenic parasitism in dogs treated with the LBMPL vaccine, suggesting the possible role of the vaccine therapy in activating the immune response directly reflecting in the splenic parasitism control.

The splenic compartment contains the crucial elements to respond effectively to the outcome of systemic infections. Consequently, this organ undergoes sequential changes in *L. infantum*-infected dogs [12]. Several studies have shown disorganization of spleen structure, and the microarchitecture disruption has been associated with disease progression. Hyperplastic reactive lymphoid follicles and increased number of macrophages in white pulp associated with increased cellularity in the red pulp are morphological alterations observed in CVL [14,19,20,21,22]. Differently, in this study, we verified a reduction in red pulp hyperplasia, reduction in follicular area and white pulp organization in the LBMPL group, demonstrating the possible role of immunotherapy in recovering the splenic microarchitecture. These results are consistent with those of Tafuri et al. [19] and Cavalcanti et al. [6] who demonstrated an increase in mononuclear cells in the red pulp and progressive architecture disruption in the white pulp in symptomatic dogs and dogs with high splenic parasitism, respectively. Associated with splenic microarchitecture disruption, spleen enlargement is a constant finding in the *L. infantum* infection [8,20]. In a previous study, Roatt et al. [17] observed a strong reduction in splenomegaly in dogs treated with the LBMPL vaccine. Reduction in spleen size is used as a parameter of therapeutic response both in human and canine VL, suggesting the impact of immunotherapy in attenuating the morphological changes in the spleen [20,23,24,25].

The splenic morphological changes are associated with a state of immunosuppression that compromises the host capacity to control the *Leishmania* infection, hindering the activation of the immune system [20]. In view of this, we investigated the cytokine mRNA expression and iNOS in the splenic compartment after LBMPL immunotherapy. Our results demonstrated an increased mRNA expression of IFN-γ, TNF-α, IL-12 and iNOS and a decrease in IL-10 and TGF-β1 in dogs treated with the LBMPL vaccine. Previous studies suggest an association between Th1 immune response and resistance in the CVL, where IFN-γ and TNF-α act synergistically in the macrophage’s activation, which through iNOS produce reactive nitrogen species [20,26]. Consistent with our data, Dos Santos et al. [27] observed an increase in iNOS in splenic tissue in symptomatic dogs, as well as a positive correlation between the number of iNOS^+^ cells and IFN-γ levels in the spleen. On the other hand, IL-10, TGF-β1, and IL-4 are associated with susceptibility profile in *L. infantum* infected dogs [20,28]. Lage et al. [29] observed an increase in IL-10 associated with an increase in splenic parasitic load and progression of the disease in CVL. Together, these data reinforce that the LBMPL vaccine therapy induces a protective immunity, with a possible activation of phagocytes and recovery of the splenic architecture. 

Associated with morphological and immunological findings, we proposed a quantification of splenic parasitism. A strong reduction in spleen parasite burden was observed in the dogs after LBMPL immunotherapy. Possibly, the activation of the immune response, with a balance to the Th1 profile, contributes to the killing of parasites. Corroborating with our data, Roatt et al. [17] observed a reduction in parasite load in bone marrow and skin after treatment with the LBMPL vaccine. Additionally, the reduction in the parasitism was associated with a recovery of splenic microarchitecture, similar to Cavalcanti et al. [6] who observed an association between low parasite load and microarchitecture organization in the spleen of *L. infantum*-infected dogs, unlike dogs with high parasitism that presented rupture of the splenic microarchitecture.

## 4. Conclusions

In conclusion, this study demonstrated an organization of splenic architecture and increased mRNA expression of pro-inflammatory cytokines, associated with reducing parasite burden in LBMPL treated dogs. Possibly, the inflammatory cytokine environment produced upon LBMPL immunotherapy causes the progressive organization of the splenic architecture favoring cell migration, antigen presentation and lymphocyte activation. Consequently, there is a widespread decline in inflammatory cytokines, with consequent parasite control leading to a strong local and tissue specific resistance. These results, associated with those previously published, provide the potential of the LBMPL vaccine as a possible therapeutic alternative in dogs infected with *L. infantum*. 

## 5. Materials and Methods

### 5.1. Animals and Ethics Statement 

Twenty-three dogs of different breeds, ages and sex, naturally infected with *L. infantum,* were obtained through donation by the Centro de Controle de zoonoses (CCZ) of Governador Valadares City (Minas Gerais State, Brazil). The infection was initially identified by serological tests (Dual-Path Platform—DPP^®^ and ELISA) performed by the CCZ as part of the activities of the municipality’s VL Control Program. Then, the infection of the animals by *L. infantum* was confirmed at the Laboratorio de Imunopatologia (UFOP) by specific quantitative PCR and culture of bone marrow aspirate samples. All animals were symptomatic, presenting at least three clinical signs related to the CVL (weight loss, dermatitis, lymphadenopathy, alopecia) [17]. As a control group, five uninfected dogs were used in this study, totaling twenty-eight animals.

All dogs were distributed in collective stalls and pre-treated with anti-tick, anti-scabies and anti-helminthic drugs and vaccinated against rabies (Tecpar, Sao Paulo, Brazil), Parainfluenza, Coronaviruses and Leptospirosis (Vanguard Plus, Pfizer, New York, NY, USA). Water and food were available ad libitum. The stalls were cleaned with 1% sodium hypochlorite daily, and the animals received recreation inside the stalls.

### 5.2. Treatment Protocols

The dogs were randomly allocated to one of the following groups in an open fashion, and treatment was started: INT group—n = 7 (dogs that no received any treatment); MPL group—n = 6 (dogs that received treatment only with monophosphoryl lipid A adjuvant); LBMPL group—n = 10 (dogs that received treatment with the vaccine composed of *L. braziliensis* promastigote protein associated with MPL adjuvant). The immunotherapeutic scheme was composed by three treatment series, each series composed of 10 doses. In the first series of treatment, dogs received increasing concentrations of the vaccine antigen as described: on day 1, dogs received 60 μg of *L. braziliensis* antigen protein + 5 μg of MPL in 1 mL of sterile 0.9% saline; day 2, dogs received 120 μg of *L. braziliensis* antigen protein + 10 μg of MPL in 1 mL of sterile 0.9% saline; day 3, dogs received 180 μg of *L. braziliensis* antigen protein + 15 μg of MPL in 1 mL of sterile 0.9% saline; day 4, dogs received 240 μg of *L. braziliensis* antigen protein + 20 μg of MPL in 1 mL of sterile 0.9% saline; day 5, dogs received 300 μg of *L. braziliensis* antigen protein + 25 μg of MPL in 1 mL of sterile 0.9% saline. From day 6 to day 10, dogs received 300 μg of *L. braziliensis* antigen protein + 25 μg of MPL in 1 mL of sterile 0.9% saline daily, and 10 days of interval were given for the second series. In the second and third series, 300 μg of *L. braziliensis* antigen protein + 25 μg of MPL in 1 mL of sterile 0.9% saline was injected via a subcutaneous route daily for 10 days, with 10 days of interval between each series [17]. A similar treatment regimen was used for the adjuvant alone (MPL group). In the first series of MPL treatment, dogs received increasing concentrations of the adjuvant as described: on day 1, dogs received 5 μg of MPL in 1 mL of sterile 0.9% saline; day 2, dogs received 10 μg of MPL in 1 mL of sterile 0.9% saline; day 3, dogs received 15 μg of MPL in 1 mL of sterile 0.9% saline; day 4, dogs received 20 μg of MPL in 1 mL of sterile 0.9% saline; day 5, dogs received 25 μg of MPL in 1 mL of sterile 0.9% saline. From day 6 to day 10, dogs received 25 μg of MPL in 1 mL of sterile 0.9% saline daily, and 10 days of interval were given for the second series. In the second and third series, 25 μg of MPL in 1 mL of sterile 0.9% saline was injected via a subcutaneous route daily for 10 days, with 10 days of interval between each series [17]. All animals were monitored daily for clinical and behavioral changes following vaccine applications. As a control group, five uninfected and untreated dogs were used in this study (NI group—n = 5).

The MPL (monophosphoryl lipid A, obtained from *Salmonella enterica*, serotype Minnesota RE 595) (Sigma Chemical Co., St. Louis, MO, USA) adjuvant used in association with the *L. braziliensis* antigen was prepared in accordance with the manufacturer’s recommendations. At the time of the injection, the MPL was diluted in sterile 0.9% saline and was mixed with the vaccine antigen to avoid loss of stability. The *L. braziliensis* antigen was prepared in accordance with Giunchetti et al. [30]. Briefly promastigotes of *L. braziliensis* (MHOM/BR/75/M2903) were maintained in in vitro culture in NNN/LIT media. Parasites were harvested via centrifugation (2000× *g*, 20 min, 4 °C) from 10-day-old cultures, washed three times in sterile saline buffer, totally disrupted by ultrasound (40 W, 1 min, 0 °C), separated into aliquots and stored at −80 °C until required for use. Protein concentration was determined according to the Lowry method. As described before, the vaccine antigen and the MPL adjuvant were diluted in sterile 0.9% saline and mixed in the moment of use to avoid loss of stability.

### 5.3. Euthanasia Protocol and Necropsy

After receiving therapeutic applications, all dogs underwent euthanasia and necropsy 90 days (T90) after treatments. For this, the animals were submitted to anesthetic protocol comprising 22 mg/kg of ketamine (Ketamina Agener®, Agener União, Brasil) combined with 2 mg/kg of xylazine hydrochloride (Calmium®, Agener União, Brasil) via an intramuscular route. Following previous anesthesia, the dogs received sodium thiopental (Thiopenta× 1 g, Cristália, Brazil) at 20 mg/kg for general anesthesia, with subsequent administration of potassium chloride for circulatory collapse. Animals were clinically examined, and a gross examination of internal organs was performed. Subsequently, the spleen was removed, and the main histopathological, immunological and parasitological alterations were analyzed.

### 5.4. Histopathology 

After euthanasia, spleen samples were collected, fixed in 10% buffered formalin solution, dehydrated, cleared, embedded in paraffin, cut into 4 μm thick sections and stained by hematoxylin/eosin for histopathological analysis. The red pulp hyperplasia was evaluated by semiquantitative analysis using optical microscopy (Olympus Optical, Japão) at 40× magnification and classified according to the degree of intensity, as follows: absent (-), mild (+), moderate (++) or intense (+++). In addition, the follicle area was quantified in total spleen demonstrated on the histological slide at 5× magnification. The images were digitalized by AxioCam MRc microcamera associated with Leica Axio Imager.Z2 microscopy. The analysis was performed using Leica Qwin V3 software (Wetzlar, Germany), quantifying the entire follicular area.

### 5.5. RNA Extraction and Quantification of the Gene Expression of iNOS and Cytokines (IFN-γ, TNF-α, IL-12, IL-10 and TGF-β1) by qRT-PCR

Total RNA was extracted from spleen tissue fragments using Trizol reagent (Invitrogen, Carlsbad, CA, USA), followed by final purification with SV Total RNA Isolation System kit (Promega, Madison, WI, USA) in accordance with the manufacturer’s recommendation. RNA quantification was performed using Nanodrop spectrophotometer (NanoDrop Lite spectrophotometer; Thermo Fisher Scientific, Carlsbad, CA, USA). From 1 μg of RNA, the cDNA synthesis was performed using a High-Capacity cDNA Synthesis kit (Thermo Fisher Scientific, Carlsbad, CA, USA), following the manufacturer’s protocol. qPCR was performed using the SYBR^®^ Green PCR Master Mix (Applied Biosystems/ Thermo Fisher Scientific, Carlsbad, CA, USA) on a 7500 Real-Time PCR System (Applied Biosystems, Carlsbad, CA, USA). Primers used to amplify cytokines (IFN-γ, TNF-α, IL-12, IL-10, TGF-β1) and iNOS genes were designed with the aid of Gene Runner version 6.0 using specific canine sequences obtained from GenBank (http://www.ncbi.nlm.nih.gov/genbank/ accessed on 18 November 2021) according to Menezes-Souza et al. [18]. GAPDH was used as reference for normalization of expression of the genes of interest. The results were expressed according to the 2^-ΔΔCt^ method. 

### 5.6. Splenic DNA Extraction and Quantification of Parasite Load by Quantitative PCR

Total genomic DNA was extracted from approximately 20 mg of spleen samples using Wizard SV Genomic DNA Purification System Kit (Promega Corporation, Madison, WI, USA) in accordance with manufacturer recommendations. Briefly, 20 mg of spleen samples was incubated with Proteinase K digestion solution in a 55 °C heat block overnight. After that, 250 μL of Wizard® SV Lysis Buffer was added to each sample, vortex and the lysate was transferred to Wizard® SV Minicolumn. After that, a vacuum was applied until the lysate passed through the column. The column was washed with 800 μL of column wash solution. Afterwards, the minicolumn/tube was assembled into the centrifuge and spun at 13,000× *g* for 1 min for DNA elution. Good laboratory practice was used to avoid DNA cross-contamination, and negative controls were included during all DNA extraction procedures. The concentration and purity of DNA was determined using NanoDrop^®^ spectrophotometer (Thermo Fisher Scientific, Carlsbad, CA, USA) according to A_260_/A_280_ and A_260_/A_230_ analysis.

To quantify spleen parasite burden, primers were used that amplified a 90 bp fragment of a single copy of DNA polymerase gene of *L. infantum* (GenBank accession number AF009147). The integrity of spleen samples was analyzed using GAPDH primers (AB038240) that amplified a 115 bp fragment. Reactions were processed in duplicate. Standard curves were prepared for each run using known quantities of *L. infantum* promastigotes (OP46 strain). For this, purified DNA was diluted 1:10 (from 10^6^ until 1.0 parasite/μL). Each curve point was performed in triplicate. The DNA amplification was performed using 1× TaqMan Universal Master Mix (Applied Biosystems, Carlsbad, CA, USA), 20 pmol of the primers, 10 pmol of the probe and 20 ng of total DNA. The run was conducted in a 7500 Real-Time PCR System thermocycler (Applied Biosystems, Carlsbad, CA, USA). The result was expressed as the number of amastigotes DNA copies per mg of the spleen. 

### 5.7. Statistical Analysis

GraphPad Prism software—version 8.0 (Prism Software, San Diego, CA, USA) was used for statistical analysis. The Kolmogorov–Smirnov test was used to assess the normal distribution of the data. The analyses were conducted using one-way ANOVA for variables with parametric distributions and the Kruskal–Wallis test for variables with nonparametric distributions. Analysis of red pulp hyperplasia was carried out using the Chi-square test. Correlations were determined using Pearson rank correlation coefficient. In all cases, *p*-value *<* 0.05 was considered significant.

## Figures and Tables

**Figure 1 pathogens-11-00974-f001:**
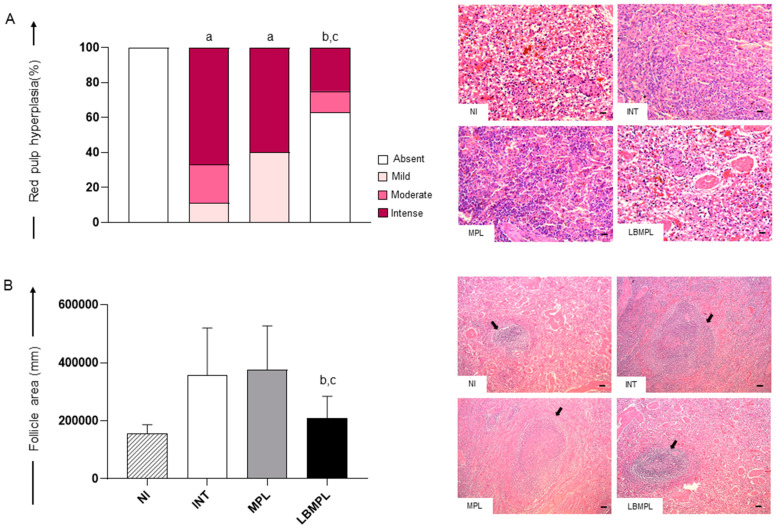
Histological changes in the spleen of dogs uninfected and untreated (NI), dogs infected by *L. infantum* and untreated (INT), dogs infected by *L. infantum* and treated with the monophosphoryl lipid A (MPL) adjuvant alone and dogs infected by *L. infantum* and treated with the vaccine composed of *L. braziliensis* antigens plus MPL adjuvant (LBMPL). (**A**) Red pulp evaluation demonstrating the presence or absence of hyperplasia, expressed in percentage as follows: white bar = absent hyplerplasia, light pink = mild hyplerplasia, pink = moderate hyplerplasia, fuchsia = intense hyperplasia. Representative photomicrographs of different degrees of hyplerplasia of red pulp: normal structure of red pulp in NI group; intense hyperplasia of red pulp in INT and MPL groups; and absent hyperplasia in LBMPL group. Hematoxylin–eosin staining was used. Bar = 100 μm. Images shown at 20× magnification. (**B**) White pulp evaluation demonstrating the area occupied by lymphoid follicles. The results are expressed as mean ± standard deviation of follicle area in millimeters. Representative photomicrographs of splenic follicles (arrows) demonstrating the organized splenic white pulp in NI and LBMPL groups, and disorganized splenic white pulp in INT and MPL groups. Hematoxylin–eosin staining was used. Bar = 400 μm. Images shown at 5× magnification. The significant differences (*p* < 0.05) are represented by “a”, “b”, and “c” referring NI, INT, and MPL groups, respectively.

**Figure 2 pathogens-11-00974-f002:**
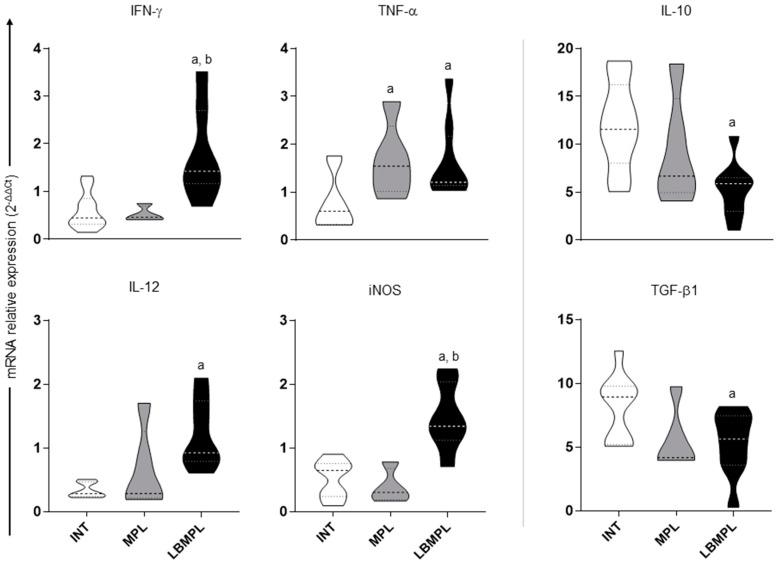
Gene expression of cytokines and iNOS in the spleen of dogs infected by *L. infantum* and untreated (INT), dogs infected by *L. infantum* and treated with the monophosphoryl lipid A (MPL) adjuvant alone and dogs infected by *L. infantum* and treated with the vaccine composed of *L. braziliensis* antigens plus MPL adjuvant (LBMPL). Analysis by qPCR of the mRNA expression of IFN-γ, TNF-α, IL-12, iNOS, IL-10, and TGF-β1 in the spleen of dogs was performed according [18]. Gene expression values were normalized according to the constitutive GAPDH gene. The significant differences (*p* < 0.05) are represented by “a” and “b” referring to INT and MPL groups.

**Figure 3 pathogens-11-00974-f003:**
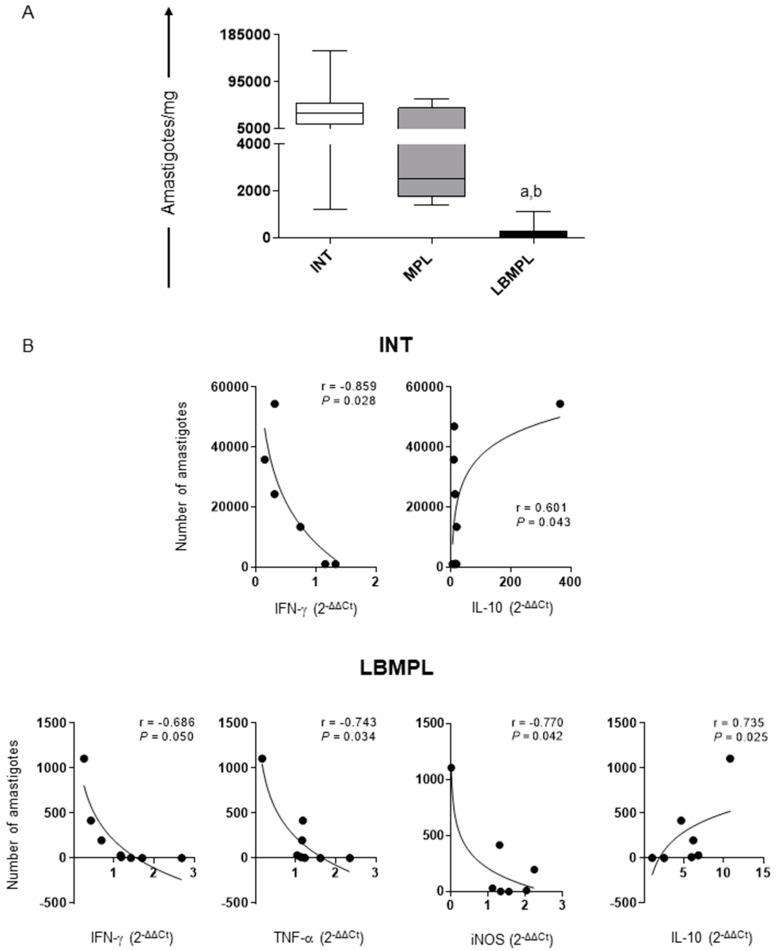
Parasite burden and cytokines correlation in the spleen of dogs infected by *L. infantum* and untreated (INT), dogs infected by *L. infantum* and treated with the monophosphoryl lipid A (MPL) adjuvant alone and dogs infected by *L. infantum* and treated with the vaccine composed of *L. braziliensis* antigens plus MPL adjuvant (LBMPL). (**A**) Splenic parasite burden was assessed through quantitative PCR, and results are expressed in amastigote DNA copies per milligram of spleen. Significant differences (*p* < 0.05) are represented by “a” and “b” referring INT and MPL groups, respectively. (**B**) The main correlation analysis between splenic parasite load and cytokines/iNOS expression are showed by Pearson’s r. Connecting lines illustrated positive and negative correlation indexes.

## Data Availability

The data set will be made available on reasonable request to the study principal investigator (roatt@ufop.edu.br).

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
