# Peer review of "LBMPL Vaccine Therapy Induces Progressive Organization of the Spleen Microarchitecture, Improved Th1 Adaptative Immune Response and Control of Parasitism in *Leishmania infantum* Naturally Infected Dogs"

_pathogens, 2022, doi:10.3390/pathogens11090974_

Round 1

Reviewer 1 Report

The manuscript by Roatt shows that Leishmania-infected dogs treated with the LBMPL vaccine exhibit low parasitism and increase the Th1 immune response. 

Although interesting, some data should be added before acceptance.

1) Visceral leishmaniasis is a chronic infection, and parasites can be found in the spleen, liver, bone marrow, lymphoid organs as well as other organs. So, to test whether a given vaccine is effective in a systemic disease, the parasite load of all classical affected organs should be analyzed; otherwise it is not possible to classify this formulation as therapeutic. At least an immunopathological evaluation of the liver should be performed.

2) Some major points must be clarified:
A) Page 2, lines 88-91: it is not possible to observe macrophage hyperplasia in figure 1A. The authors must add scale to all images. In graph 1, the color in the small square representing "moderate" doesn't match with the color of respective bars.

B) In the legend of figure 2, what does the letter "a" mean in comparison with INT (infected)? and "b"with MPL? If so, add, respectively. 

C) In the subsection 5.2 I couldn't understand the treatment scheme. How long did the animals receive 30 mg of antigen? What about MPL? When was it changed to 300ug? At this time point, how much MPL was injected? Did you inject an intermediate dose between 30-300ug? Did you inject 30 mg of vaccine every day, for 5 consecutive days? I did not follow the rationale for the second and third series. I believe the authors must detail such treatment.

D) Details the origin of the MPL adjuvant. Additionally, how did you produce the LB vaccine? How was it formulated to MPL? 

E) Itens 5.6 and 5.7 - must be split in one section. 

F) What immunoreactive antigens exist in such vaccine? Is the vaccine produce with whole antigen of L. braziliensis? What's the advantages of using such generation of vaccines instead a well defined immunogen?

Reviewer 2 Report

Excellent work. There is no plagiarism and it is very well understood. I only have a few comments:

1. Check that the whole document has the same font, for example in section 5.7 the word L. infantum is in a different font and size

2. Was the methodology described in section 5.7 adapted from another author?

3. I consider that in section 5.6 the DNA extraction process should be explained in a brief summary.

4. Why do you carry out the experiments infecting with L. infantum? dogs are usually affected by L. tropica or L. major

Round 2

Reviewer 1 Report

The work improved a lot, however, the scheme of treatment needs to be rewritten.

For example: on day, animals received [A] ug of L.b whole antigen plus [B] of MPL; on day 2 [C] ug of L.b whole antigen plus [D] of MPL, etc. If the authors will not wish to detail this scheme in the text, at least add a table containing how the whole treatment was performed.

I am asking that because if other groups want to replicate this experiment, it will not be possible because details are missing.

Author Response

Dear Reviewer#1,

As Suggested we rewrite and improve the description of the treatment scheme as follows:

In the first series of treatment, dogs received increasing concentrations of the vaccine antigen as described: day 1, dogs received 60 μg of L. braziliensis antigen protein + 5 μg of MPL in 1 mL of sterile 0.9% saline; day 2, dogs received 120 μg of L. braziliensis antigen protein + 10 μg of MPL in 1 mL of sterile 0.9% saline; day 3, dogs received 180 μg of L. braziliensis antigen protein + 15 μg of MPL in 1 mL of sterile 0.9% saline; day 4, dogs received 240 μg of L. braziliensis antigen protein + 20 μg of MPL in 1 mL of sterile 0.9% saline; day 5, dogs received 300 μg of L. braziliensis antigen protein + 25 μg of MPL in 1 mL of sterile 0.9% saline. From day 6 to day 10, dogs received 300 μg of L. braziliensis antigen protein + 25 μg of MPL in 1 mL of sterile 0.9% saline, daily and 10 days of interval was given for the second series. In second and third series, 300 μg of L. braziliensis antigen protein + 25 μg of MPL in 1 mL of sterile 0.9% saline was injected by subcutaneous route daily for 10-days with 10 days of interval between each series [17]. A similar treatment regimen was used for the adjuvant alone (MPL group). In the first series of MPL treatment, dogs received increasing concentrations of the adjuvant as described: day 1, dogs received 5 μg of MPL in 1 mL of sterile 0.9% saline; day 2, dogs received 10 μg of MPL in 1 mL of sterile 0.9% saline; day 3, dogs received 15 μg of MPL in 1 mL of sterile 0.9% saline; day 4, dogs received 20 μg of MPL in 1 mL of sterile 0.9% saline; day 5, dogs received 25 μg of MPL in 1 mL of sterile 0.9% saline. From day 6 to day 10, dogs received 25 μg of MPL in 1 mL of sterile 0.9% saline, daily and 10 days of interval was given for the second series. In second and third series, 25 μg of MPL in 1 mL of sterile 0.9% saline was injected by subcutaneous route daily for 10-days with 10 days of interval between each series [17].

We believe that the requested changes and the contributions of the Reviewer#1 have significantly improved the quality and understanding of our manuscript. We would like to thank the Pathogens members for your attention in revising this article and hope that it is now compatible with the high quality of Pathogens prints and that the revised version is now acceptable for publication.